# Are neural scaling laws leading quantum chemistry astray?

## Abstract

Neural scaling laws are driving the machine learning community toward training ever-larger foundation models across domains, assuring high accuracy and transferable representations for extrapolative tasks. We test this promise in quantum chemistry by scaling model capacity and training data from quantum chemical calculations. As a generalization task, we evaluate the resulting models' predictions of the bond dissociation energy of neutral $H_2$, the simplest possible molecule. We find that, regardless of dataset size or model capacity, models trained only on stable structures fail dramatically to even qualitatively reproduce the $H_2$ energy curve. Only when compressed and stretched geometries are explicitly included in training do the predictions roughly resemble the correct shape. Nonetheless, the largest foundation models trained on the largest and most diverse datasets containing dissociating diatomics exhibit serious failures on simple diatomic molecules. Most strikingly, they cannot reproduce the trivial repulsive energy curve of two bare protons, revealing their failure to learn the basic Coulomb's law involved in electronic structure theory. These results suggest that scaling alone is insufficient for building reliable quantum chemical models.

## 1 Introduction

What makes a physics model "good"? At its essence, any model is, by construction, a simplification of complex reality. Yet, a useful one would ideally reflect our current understanding of physics by incorporating as many known first-principle laws as possible while applying Occam's razor to reproduce the correct empirical observations *for the correct reasons*. For example, in quantum chemistry, Kohn-Sham density functional theory (Hohenberg & Kohn, 1964; Kohn & Sham, 1965) is an exact theory that states the existence of an exact universal functional that yields the exact energy of a given electron density. In practice, this exact functional's closed-form analytical expression is unknown and must be approximated. Encouragingly, it has been observed that non-empirical functionals *without any fitted parameters* tend to not only become more accurate as they satisfy more known exact constraints on the hierarchy of "Jacob's ladder" of density functional approximations but are also more accurate and generalizable than parameterized counterparts (Perdew & Schmidt, 2001; Medvedev et al., 2017; Goerigk & Grimme, 2011; Kaplan et al., 2023; Khan, 2025).

In contrast, machine learning (ML) models attempt to represent physical interactions from data, often by learning a direct statistical mapping from input to labeled outputs. Therefore, the central premise of using ML models to predict chemical properties is that their inference can be accurate while being orders-of-magnitude faster than *ab initio* methods that solve the Schrödinger equation (Rupp et al., 2012). The ML community, particularly the deep learning subfield, has also broadly observed across several domains encompassing *e.g.* natural language processing, computer vision, protein structure prediction, and chemistry that models trained with larger amounts of data, compute, and model parameters systematically improve performance in a power-law fashion (Cortes et al., 1993; Bahri et al., 2024; Hestness et al., 2017; Kaplan et al., 2020; Alabdulmohsin et al., 2022; Jumper et al., 2021; Dubey et al., 2024; Hoffmann et al., 2022; Cheng et al., 2024; Frey et al., 2023; Jiang et al., 2025; Hattori et al., 2025). Such "neural scaling laws" are encouraging a paradigm toward the development of large "foundation models" trained on vast amounts of diverse data (Bommasani, 2021). Accordingly, these models can demand enormous amounts of monetary, natural, and compute resources in their training (Cottier et al., 2024; Hao, 2025) but are claimed to have produced meaningful representations with strong generalization capabilities, as implicated

by universal approximation theorems (Augustine, 2024). However, it remains unclear whether large-scale quantum chemical foundation models can truly learn essential physics required to reliably generalize to novel molecules and materials beyond their training set.

In this work, we investigate whether large neural networks trained on quantum chemical data can indeed capture the governing physics needed for reliable generalization, by scaling models on two large datasets of equilibrium-geometry molecules. Our scaling experiments agree with the literature that increasing the number of training samples improves performance — at least, on the hold-out test split. We find that irrespective of training data quantity and model capacity, there is no discernible improvement in the models' reproductions of the bond dissociation energy (BDE) curve of $H_2$, the smallest and simplest possible molecule. The models improve modestly only when non-ground-state structures are included in the training set. However, even the largest foundation models trained on over 101M structures that include dissociating bonds can fail to qualitatively describe the BDE curves of $H_2$ and other diatomic molecules. More disconcertingly, they cannot reproduce the trivial energy curve for a system of two bare protons produced from the fundamental and classical Coulomb's law. Contrary to the current pervasive inclination in the ML community to favor scaling, our results indicate that training ever-larger models on ever-larger datasets does not necessarily lead to "good" quantum chemical models that give correct answers for the correct reasons.

## 2 BACKGROUND

We provide a brief overview of the following topics that pertain to our experiments described in section 3: theoretical quantum chemistry, ML applications in quantum chemistry, neural scaling laws, and foundation models.

### 2.1 QUANTUM CHEMISTRY

Quantum chemistry uses quantum mechanical principles to accurately calculate properties of an arbitrary chemical system composed of electrons, protons, and neutrons. This requires solving the Schrödinger equation (Schrödinger, 1926), which for $N$ electrons is a linear partial differential equation, given in its time-independent form as

$$\hat{H}\Psi(\mathbf{r}_1, \ldots, \mathbf{r}_N) = E\Psi(\mathbf{r}_1, \ldots, \mathbf{r}_N) \tag{1}$$

where $\Psi(\mathbf{r}_1, \ldots, \mathbf{r}_N)$ is the antisymmetric and normalized wavefunction describing the state, $E$ is the total energy, and each $\mathbf{r}_i$ denotes the spatial-spin coordinate of the $i$th electron. In atomic units, the non-relativistic Hamiltonian operator, $\hat{H}$, corresponds to $E$ of $\Psi$ and is defined under the Born-Oppenheimer approximation (Born & Oppenheimer, 1927) of stationary nuclei as

$$\hat{H} = -\sum_i \frac{1}{2}\nabla^2_{\mathbf{r}_i} - \sum_i \sum_j \frac{\mathcal{Z}_j}{|\mathbf{r}_i - \mathbf{R}_j|} + \sum_i \sum_{k>i} \frac{1}{|\mathbf{r}_i - \mathbf{r}_k|} + \sum_j \sum_{l>j} \frac{\mathcal{Z}_j \mathcal{Z}_l}{|\mathbf{R}_j - \mathbf{R}_l|} \tag{2}$$

where $\mathbf{R}_j$ is the position of the $j$th nucleus and $\mathcal{Z}_j$ is its nuclear charge. From left to right, the terms respectively represent the electronic kinetic energy, the potential energy from electron-nucleus interactions, the potential energy from electron-electron repulsions, and the potential energy from nucleus-nucleus repulsions. Unfortunately, a closed-form solution to the Schrödinger equation in its differential form does not exist for multi-electron systems due to the electronic repulsion term (Tew et al., 2007). Moreover, the exponential increase with $N$ in the dimension of the Hilbert space spanned by the different electronic configurations (Kohn, 1999) necessitates alternative solution methods that are computationally tractable.

### 2.1.1 KOHN-SHAM DENSITY FUNCTIONAL THEORY

One such widely-employed method (Jones, 2015) is density functional theory (DFT). It is an exact theory rooted in the universal Hohenberg-Kohn theorems (Hohenberg & Kohn, 1964) that concerns any system of electrons in any external potential:

- **Theorem 1**: To within an additive constant, the external potential, and thereby the total system energy is a unique functional of the ground-state electron density, $n$.

- **Theorem 2**: From the set of $v$-representable trial densities (Levy, 1979) associated with antisymmetric ground-state wavefunctions that are non-negative everywhere and normalize to $N$, the ground-state energy of the system is minimized at its ground-state density, $n_0$.

Consequently, DFT dramatically reduces the concerned configuration space to simply three spatial dimensions that define $n$. Then, by introducing single-electron orbitals through an auxiliary non-interacting system that exactly reproduces $n_0$ (Kohn & Sham, 1965) to solve the non-interacting electronic kinetic energy, one has Kohn-Sham DFT (KSDFT). Attractively, KSDFT requires just the exchange-correlation functional to be approximated, which comprises just a small percentage of $E$. It is also relatively inexpensive compared to wavefunction-based methods as its computational time complexity is formally at least $\mathcal{O}(N^3)$ due to orthonormalization of the orbitals and diagonalization of the Kohn-Sham Hamiltonian matrix (Lee & Dieng, 2025). However, its accuracy can vary widely depending on the utilized functional (Goerigk & Grimme, 2011).

### 2.1.2 G4(MP2) THEORY

Gaussian-4(MP2) theory (Pople et al., 1989; Curtiss et al., 2011; 2007b;a), G4(MP2), is a compositve wavefunction method (Helgaker et al., 2008). It can robustly predict energies to within chemical accuracy of 1 kcal/mol relative to experimental data for molecules containing first-, second-, and third-row main-group elements. It achieves this by performing several calculations at different levels of theory, including: Hartree-Fock (Slater, 1930) energies extrapolated to the complete basis set limit; coupled cluster singles and doubles perturbative triples (CCSD(T)) (Csirik & Laestadius, 2023) energies calculated with the 6-31G* basis set (Rassolov et al., 1998); and frozen-core energy corrections calculated with second-order Møller-Plesset perturbation theory (MP2) (Møller & Plesset, 1934) using the G3MP2LargeXP (Curtiss et al., 2007b) and 6-31G(d) basis sets (Rassolov et al., 1998).

### 2.2 MACHINE LEARNING IN QUANTUM CHEMISTRY

Although KSDFT and G4(MP2) are highly-accurate, their computational time complexity of $\mathcal{O}(N^3)$ or worse (*e.g.* CCSD(T) scales as $\mathcal{O}(N^7)$ (Ratcliff et al., 2017)) means *ab initio* quantum chemical calculations are prohibitively expensive for even small molecules. Therefore, ML has been applied extensively to quantum chemistry to bypass the computational barrier of repeatedly having to explicitly solve the Schrödinger equation for each new molecule and material. In other words, ML models can learn a statistical mapping from the external potential defined by the sets of nuclear charges, $\{\mathcal{Z}_j\}$, and nuclear coordinates, $\{\mathbf{R}_j\}$, to any given property (Rupp et al., 2012). This is akin to the universal functional stated in Hohenberg and Kohn's first theorem. Of course, the important distinction is that these ML models can be orders-of-magnitude faster than traditional quantum chemical methods without much loss in accuracy. We refer the reader to many excellent references that comprehensively review the numerous ML applications in chemistry (Schütt et al., 2020; Westermayr et al., 2021; Keith et al., 2021; Sajjan et al., 2022; Huang et al., 2023; Huang & Von Lilienfeld, 2021; von Lilienfeld & Burke, 2020; Von Lilienfeld, 2018; Hansen et al., 2013; von Lilienfeld et al., 2020; Dral, 2020; Malica et al., 2025).

### 2.2.1 NEURAL SCALING LAWS

The accuracy of a ML model's predictions has been empirically observed to depend on the quality and quantity of the training data, and the employed learning algorithm. As mentioned in section 1, for neural network architectures, neural scaling laws refer to observations that more training data and larger models reduce prediction errors as a power-law. For example, Frey et al. (2023) showed that state-of-the-art graph neural network architectures like SchNet (Schütt et al., 2017), PaiNN (Schütt et al., 2021), Allegro (Musaelian et al., 2023), and SpookyNet (Unke et al., 2021a) tend to improve their predictions of atomic forces with increasing model capacity, $c$, defined as

$$c = d \times w \tag{3}$$

where $d$ is the number of convolutional layers and $w$ is the embedding dimension. Moreover, they show monotonic improvements to the loss with increasing dataset size.

### 2.2.2 FOUNDATION MODELS

Neural scaling laws are motivating the development of foundation models (Bommasani, 2021), which can be broadly defined as large models trained on large and diverse datasets that can subsequently be fine-tuned for a wide range of downstream tasks. Examples include BERT (Devlin et al., 2019), GPT-3 (Brown et al., 2020), and CLIP (Radford et al., 2021) for text- and image-related tasks. The assumption is that they have learned general representations that can offer potential solutions to data-scarce and extrapolative regimes. Unsurprisingly, this is appealing for chemical applications and many chemical foundation models have accordingly been developed (Yuan et al., 2025; Choi et al., 2025; Pyzer-Knapp et al., 2025; Alampara et al., 2025; Ahmad et al., 2022; Batatia et al., 2023; Chen & Ong, 2022; Unke et al., 2021b; Wood et al., 2025; Soares et al., 2024; Beaini et al., 2023; Merchant et al., 2023; Yang et al., 2024).

## 3 EXPERIMENTS

Here, we are interested in whether neural scaling laws and foundation models indeed produce "good" quantum chemical models by having learned general and physically-meaningful representations that would allow for these large models to be reliably applied to novel molecules and materials.

### 3.1 $H_2$ AS A FUNDAMENTAL TEST

To investigate this, we consider whether scaling deep neural networks trained on molecules in their equilibrium geometries can lead to accurate predictions of the bond dissociation energy (BDE) curve at different bond lengths, $R$, of neutral $H_2$ — the smallest and simplest possible molecule consisting of only two protons, two electrons, and one bond. The BDE curve for a diatomic plots the total energy versus bond length, $R$, normalized by subtracting the summed energies of the two isolated atoms from the total system energy. This offers a simple evaluation of the generalization capabilities of the models since there are no complications arising from functional groups, conformational degrees of freedom, relativistic effects (Pyykkö, 2012), unexpected spin multiplicities, core electrons, dipole moments, or the need for large basis sets (Boese et al., 2003).

However, the BDE curve of $H_2$ is still challenging to accurately describe because of the strong-correlation limit approached in the exchange-correlation energy as the bond dissociates, at which each electron should be localized to their individual nuclei. This regime is accurately described as a multi-reference system where many Slater determinants contribute to the exact wave function. Consequently, single-determinant methods like KSDFT, which assumes the electron density can be described by one Slater determinant, would often be inadequate to provide quantitatively accurate estimates of the $H_2$ BDE. Therefore, it may perhaps be unrealistic for the trained models to exactly reproduce the exact BDE curve. Nonetheless, we expect that they are capable of reproducing the BDE curve to the accuracy of the level of theory used to generate the training data. Further, we maintain that a model that has properly learned the basic physics involved, as may be anticipated from neural scaling laws, should capture some essential features: no kinks and divergent behavior in the BDE curve; an asymptotic plateau in the energy as the inter-nuclear distance increases; and the obvious limit of $\lim_{R \to 0} E \to +\infty$ as the nuclear-nuclear repulsion energy, $E_{nn}$, dominates and diverges due to the inverse-$R$ relation given by

$$E_{nn} = \frac{\mathcal{Z}_j \mathcal{Z}_l}{R} \tag{4}$$

In our work, we obtain reference curves from spin-restricted calculations performed using `PySCF` (Sun et al., 2018).

### 3.2 SCALING

Here, we describe our scaling experiments in which SchNet models are trained on large, high-quality quantum chemical datasets. Our objective is to systematically explore how training models of larger capacity on larger datasets affects predictive accuracy of predicting molecular energies.

### 3.2.1 SCHNET

We perform the scaling experiments by training SchNet models, implemented in `PyTorch Geometric` (Fey & Lenssen, 2019; pyt, a;b), of various $c$ by taking combinations of $d \in \{3, 6\}$ and $w \in \{100, 250, 500\}$. For all models, the number of hidden channels is set to $w$, 50 Gaussians are used to expand the pairwise distances, the cutoff distance for the interatomic interactions is 5.0 Å, and each node collects at most 32 neighbors within this cutoff. Each model is trained on several training set sizes, $N_{\text{train}} \in \{10^x, 5 \cdot 10^x \mid x \geq 2, x \in \mathbb{Z}\}$, where $x$ is capped by the dataset size. For $x \leq 3$, the batch size is set to 10 samples and for $x \geq 4$, the batch size is set to 50 samples. "Warm-up" is performed where the learning rate for a given epoch, $\epsilon$, within the first ten epochs is set to $\epsilon/10$. After this warm-up period, the learning rate is decreased to and kept fixed at $5 \cdot 10^{-4}$. However, if the tracked value of the lowest validation loss fails to decrease after twenty epochs, the learning rate is scaled by 0.5. If the validation loss does not decrease after 50 epochs, training is terminated. All models are trained with gradient descent using `PyTorch` (Paszke et al., 2019) for up to 1000 epochs using the AdamW optimizer (Loshchilov & Hutter, 2017) with mean squared errors as training and validation losses. We also clip gradients by their Euclidean norms with a threshold value of 1000. Each model is trained on a single node on compute clusters, utilizing one NVIDIA RTX A6000 GPU, 16 CPU cores, and 64 GB of RAM. The model parameters at the epoch where the validation loss is the smallest during training are saved as the best-performing model. 1000 samples are removed from the dataset and fixed as the test set across all $N_{\text{train}}$. For each model, 1000 samples are randomly chosen from the remaining dataset and used as the validation set.

### 3.2.2 DATASETS

We train these models in a supervised fashion where nuclear charges and coordinates are used to predict total atomization energies, $E_{\text{TAE}}$, reported in the GDB-9-G4(MP2) (Narayanan et al., 2019) and VQM24 datasets (Khan et al., 2024). $E_{\text{TAE}}$ is the difference between a molecule's $E$ and the sum of $E$ of its isolated constituent atoms:

$$E_{\text{TAE}} = \sum_j E_{\text{atom},j} - E_{\text{molecule}} \qquad (5)$$

GDB-9-G4(MP2) contains 133,296 stable closed-shell and neutral organic molecules in their equilibrium geometries containing no more than nine heavy atoms of C, N, O, and F in the GDB-9 database (Ramakrishnan et al., 2014; Ruddigkeit et al., 2012). These molecules' properties were computed using the highly-accurate G4(MP2) theory. VQM24 reports closed-shell and neutral organic and inorganic molecules with no more than five heavy atoms of C, N, O, F, Si, P, S, Cl, and Br. 835,947 converged molecules were computed at the $\omega$B97X-D3/cc-pVDZ (Chai & Head-Gordon, 2008; Dunning Jr, 1989) level of KSDFT. $\omega$B97X-D3 is a range-separated hybrid functional with D3 dispersion correction (Grimme et al., 2010) that has excellent performance in main-group thermochemistry, kinetics, and noncovalent interaction benchmarks (Goerigk & Grimme, 2011). Note that 51,072 of the molecules in the dataset converged to saddle points, as identified by vibrational frequency calculations. Such structures correspond to transition states, which are of highest potential energies along reaction coordinates and contain bonds that are partially broken or formed (Braddock et al., 2024).

For both datasets, we convert $E_{\text{TAE}}$ to units of kcal/mol and scale by $1/25$, which we find to be beneficial in training.

### 3.3 FOUNDATION MODELS FOR QUANTUM CHEMISTRY

We also examine five "foundation" machine-learned interatomic potentials that can estimate the energy of an atomic structure: UMA-S-1.1 (Wood et al., 2025), UMA-M-1.1 (Wood et al., 2025), OMol25 eSEN-sm-cons. (Levine et al., 2025), Orb v3 conservative OMol25 (Rhodes et al., 2025; git), and AIMNet2 (Anstine et al., 2025). As of September 2025, Rowan Benchmarks (Rowan) reported that these five models were the most accurate ML models in predicting molecular energies from $\{\mathcal{Z}_j\}$ and $\{\mathbf{R}_j\}$ on tasks involving thermochemistry, kinetics, and non-covalent interactions while being more accurate than popular density functionals, composite density functional methods, and semi-empirical methods. UMA-S-1.1 (150M parameters), UMA-M-1.1 (1.4B parameters), OMol25 eSEN-sm-cons. (6.3M parameters), and Orb v3 conservative OMol25 (25M parameters) were all trained on the OMol25 dataset (Levine et al., 2025), which contains over 101M

$\omega$B97M-V/def2-TZVPD (Mardirossian & Head-Gordon, 2016; Rappoport & Furche, 2010; Hellweg & Rappoport, 2015) KSDFT calculations that required at least 6 billion CPU hours to compute. OMol25 includes small molecules, biomolecules, metal complexes, and electrolytes that span 83 elements, a wide range of intra- and inter-molecular interactions, conformers, reactive structures, spin multiplicities, and charges. AIMNet2 supports H, B, C, N, O, F, Si, P, S, Cl, As, Se, Br, I, and can handle charged molecules. Its model size is not publicly reported but it was trained on $2 \cdot 10^7$ $\omega$B97M-D3/def2-TZVPP KSDFT calculations of bioactive molecules and small organic molecules sourced from ChEMBL (Mendez et al., 2019), PubChem (Kim et al., 2023), ANI-2x (Devereux et al., 2020), and OrbNet (Qiao et al., 2020) datasets.

### 3.4 Scaling does not necessarily help $H_2$

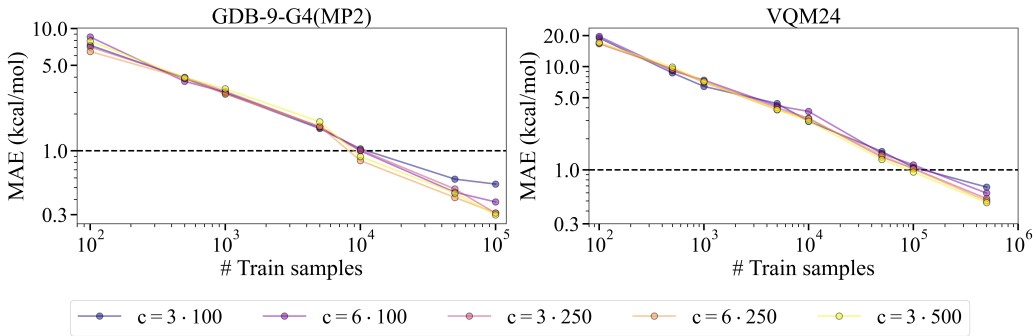

Figure 1: Prediction errors on test set splits of SchNet models of varying model capacity, $c$, trained on different numbers of training samples from (**left**) GDB-9-G4(MP2) and (**right**) VQM24 datasets. The mean absolute error is plotted against the number of training samples on a log-log plot. Horizontal dashed line denotes baseline accuracy of 1 kcal/mol.

As seen in Figure 1, we find that training SchNet models on both the GDB-9-G4(MP2) and VQM24 datasets on increasingly larger numbers of training samples systematically reduces all models' test set prediction errors. The mean absolute error decreases linearly on the log-log plot plotting error against number of training samples, as expected from prior works in the literature regarding scaling. In addition, models of larger values of $c$ tend to show lower errors, which is also not unsurprising.

What is more interesting is that the SchNet models trained on GDB-9-G4(MP2) show no systematic improvement in their abilities to even qualitatively reproduce the $H_2$ BDE curve. As shown in Figure 2 (top, left), all models trained on 100k samples predict essentially the same BDE curve, regardless of $c$. Near the bond length of 0.77 Å, the $\omega$B97M-V/cc-pVQZ KSDFT (Dunning Jr, 1989) and "exact" all-electron CCSD(T) (cc-pVQZ basis) calculations identify a minimum in BDE. However, the SchNet models rather predict curves that are practically completely horizontal and incorrectly peak to their maximum energies near 0.77 Å, despite having trained on molecules calculated using the highly-accurate G4(MP2) method. This immediately suggests an inability of scaling to allow the models trained on stable molecules to produce meaningful representations that allow for generalization to simple bond stretching. This is further apparent as the models do not obviously become more accurate with more data or greater model capacity in predicting either the equilibrium bond length (Figure 2, top, middle) or bond dissociation energy (Figure 2, top, right).

In contrast, the models trained using VQM24 exhibit improved BDE curves (Figure 2, bottom, left). Although they show many imperfections such as incorrect asymptotic behavior in energy as $R \to 0$, many kinks, and too large of a curvature near the equilibrium bond length, they are at least able to exhibit some resemblance to the reference curve of $\omega$B97M-V. There is still no clear and systematic improvement with more training data in the models' abilities to predict the equilibrium bond lengths (Figure 2, bottom, middle) and dissociation energies (Figure 2, bottom, right), but they are able to produce results reasonably close to the KSDFT baseline after training on 500k samples. This improvement relative to the models trained on GDB-9-G4(MP2) appears to arise from the saddle-point structures contained in VQM24. For comparison, GDB-9-G4(MP2) only contains stable, non-saddle-point structures without any diatomic molecules; its smallest molecules are $H_2O$ and HCN. Moreover, as seen in Figure 3, the smallest inter-atomic distance found in each

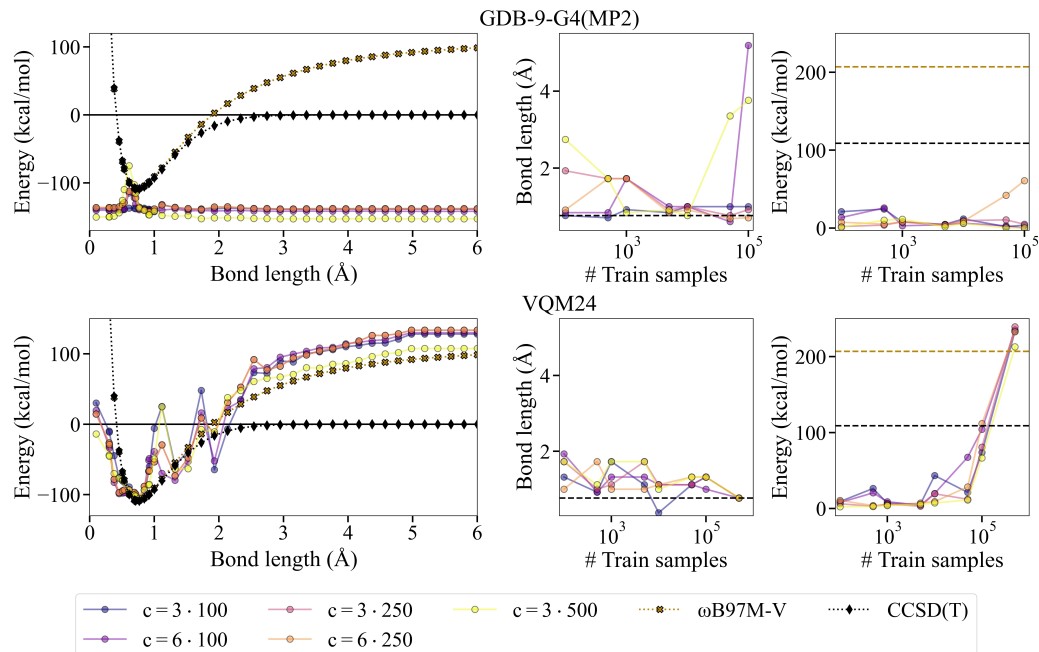

Figure 2: Neutral $H_2$ bond dissociation energy curves (left column), equilibrium bond lengths (middle column; black dashed line indicates reference CCSD(T) and KSDFT values of 0.77 Å), and dissociation energies (right column; dashed dark and light lines indicate reference CCSD(T) and KSDFT values of 109 and 207 kcal/mol, respectively) obtained from CCSD(T) (cc-pVQZ basis), $\omega$B97M-V/cc-pVQZ KSDFT, and SchNet models of various $c$ trained on (**top**) 100k GDB-9-G4(MP2) samples and on (**bottom**) 500k VQM24 samples.

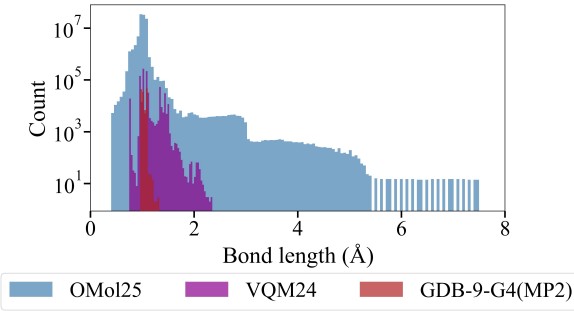

Figure 3: Distributions of smallest inter-atomic distance found in each sample in OMol25, VQM24, and GDB-9-G4(MP2) datasets.

molecule ranges between 0.96 Å and 1.33 Å with a mean of 1.03 Å. In contrast, VQM24 has more diverse bonding motifs as its smallest inter-atomic distance ranges between 0.75 Å and 2.35 Å with a mean of 1.08 Å (Figure 3). It also contains 15 diatomic molecules of HBr, HCl, HF, $N_2$, SiO, $F_2$, $O_2$, $Si_2$, BrF, ClF, $Cl_2$, $Br_2$, BrCl, PN, and $P_2$.

### 3.5 ON THE LIMITS OF FOUNDATION MODELS FOR QUANTUM CHEMISTRY

From the previous comparisons of SchNet models trained on GDB-9-G4(MP2) and VQM24 in subsection 3.4, we may expect improved performances from models trained on more diverse datasets like OMol25. Therefore, we also assess the five state-of-the-art foundation models listed in subsection 3.3 in reproducing the BDE curves of $H_2$, along with those of $Li_2$, $Be_2$, $N_2$, $O_2$, and $F_2$. Indeed, those models produce BDE curves that closely match the $\omega$B97M-V $H_2$ curve (Figure 4).

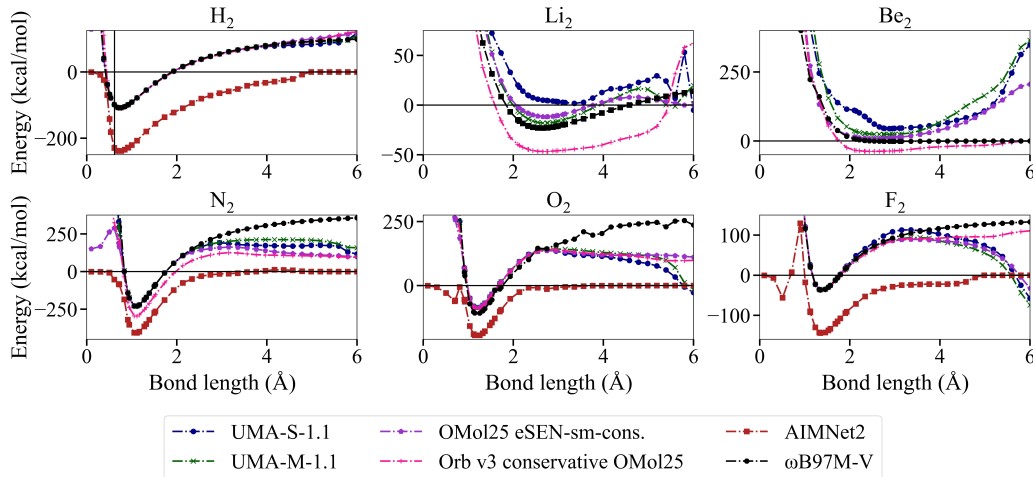

Figure 4: Bond dissociation energy curves of neutral $H_2$, $Li_2$, $Be_2$, $N_2$, $O_2$, $F_2$, calculated using $\omega$B97M-V/def2-TZVPD KSDFT and estimated using UMA-S-1.1, UMA-M-1.1, OMol25 eSEN-sm-cons., Orb v3 conservative OMol25, and AIMNet2. AIMNet2 was only applied to $H_2$, $N_2$, $O_2$, and $F_2$ because it does not support elements of the other diatomics.

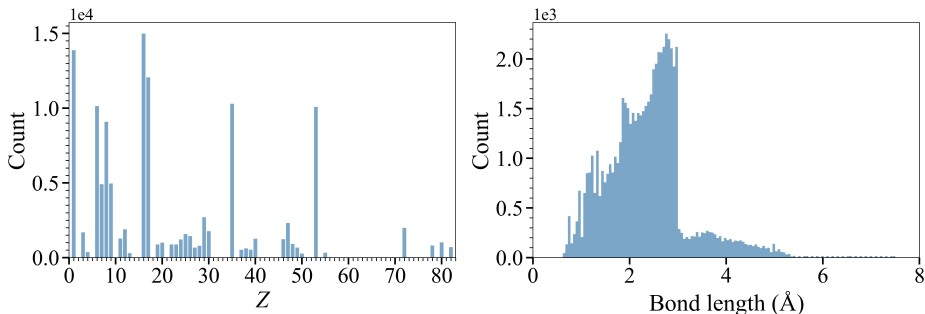

Figure 5: Distribution of (**left**) elements (denoted by nuclear charge, $\mathcal{Z}$) involved in the 61,498 diatomic systems found in OMol25 dataset and (**right**) corresponding distribution of bond lengths.

However, there is lack of transferability to other diatomic molecules as the different models show many peculiar and unphysical features, such as kinks and divergent behaviors. These limitations occur although OMol25 contains a vast assortment of molecules with compressed, equilibrium, and dissociating bonds. This is seen in Figure 3 where its distribution of smallest inter-atomic distances range from 0.40 Å and 7.50 Å, with a mean of 1.01 Å. Also, as seen in Figure 5, it encompasses 61,498 diatomic systems involving 45 different elements spanning the periodic table from H to Pb, with H, C, O, S, Cl, Br, and I being the most represented; its bond lengths range from 0.62 Å to 7.50 Å, with a mean of 2.38 Å. In fact, it contains diatomic systems involving all the elements that we have examined in Figure 4 but the models still perform poorly in accurately predicting their molecular dissociation curves. The model failures are interesting since the reference $\omega$B97M-V functional used to create these models' training data has, by construction, the correct long-range asymptotic behavior in its exchange potential that guarantees non-divergence in energies. Further, this poor performance is unexpected considering the reported energy error of $\sim$0.66 kcal/mol achieved by OMol25 eSEN-sm-cons. for bonds shorter than 6 Å (Levine et al., 2025), which is the range for the bond lengths considered in our BDE curves.

In particular, for $H_2$, $N_2$, $O_2$, and $F_2$, AIMNet2 correctly produces energies that asymptote to 0 kcal/mol for large bond lengths. This behavior may perhaps be accredited to AIMNet2's explicit calculation of dispersion energy (Grimme et al., 2010; 2016) included in a system's total energy as

$$E = E_{\text{local}} + E_{\text{dispersion}} + E_{\text{Coulomb}} \qquad (6)$$

where $E_{\text{local}}$ is the local configurational interaction energy, $E_{\text{dispersion}}$ is the dispersion interaction energy, and $E_{\text{Coulomb}}$ is the Coulomb interaction energy from partial atomic charges (Anstine et al., 2025). However, it otherwise performs very poorly as it severely underestimates relative energies at equilibrium bond lengths and wrongly plateaus in energy as $R \to 0$, despite its $E_{\text{Coulomb}}$ term.

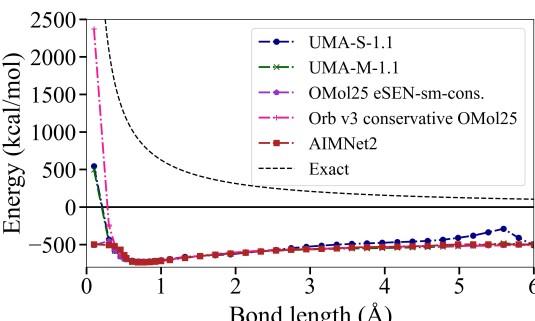

Figure 6: Total energies for a system of two bare protons as a function of inter-proton distance, estimated using UMA-S-1.1, UMA-M-1.1, OMol25 eSEN-sm-cons., Orb v3 conservative OMol25, and AIMNet2. Black dashed line indicates the exact energy profile.

We also isolate the foundation models' abilities to have learned the fundamental Coulomb's law that lies at the core of electronic structure theory by analyzing their predicted total energy curves for a system of two bare protons without any electrons. As seen in Figure 6, all of the models predict that the two-proton system is extremely stable by $\sim$500 kcal/mol even at long inter-proton distances. In fact, the predicted energy has the wrong, negative sign for practically the entire curve where the inter-proton distance is greater than $\sim$0.2 Å, whereas the exact energy should always be positive.

These results imply that scaling model capacity and training data is insufficient to attain accurate and generalizable quantum chemical property predictions. This is further concerning since even the explicit inclusion of inductive biases such as $E_{\text{Coulomb}}$ in AIMNet2 produces very poor results on the diatomic systems. These deficiencies cannot be attributed to weak model expressibility since the good results for $H_2$ in Figure 4 clearly reveal otherwise. Thus, it is not entirely obvious what fast, alternative ML methods can achieve this objective. Yet, it may be worthwhile to consider techniques like $\Delta$-ML (Ramakrishnan et al., 2015) to correct relatively-cheap semi-empirical quantum chemistry methods, and "similarity-based learning" to select minimal amounts of data in training models on-the-fly in data-scarce scenarios (Lemm et al., 2023; Lee et al., 2025).

## 4    CONCLUSION

Our experimental findings suggest that current deep neural network approaches rooted in scaling face significant challenges in producing accurate and generalizable quantum chemical models, even when trained on large, chemically diverse datasets of high-quality calculations. This limitation is evident in the bond dissociation energy curves for $H_2$, which are predicted to be essentially flat across all bond lengths by SchNet models trained on the GDB-9-G4(MP2) dataset. Training on the VQM24 dataset, which comprises off-equilibrium bonding, modestly improves reproduction of the $H_2$ curves. However, this improvement appears to stem from exposure to dissociated structures rather than from learning underlying physical principles. This is further corroborated by the observation of persistent issues in the predictions of state-of-the-art machine-learned interatomic potentials. Despite being trained on the most diverse collection of elements, chemistry, and structures to date — 101M KSDFT calculations, including 61k covering diatomic systems — their predicted energy curves for simple diatomic molecules degrade significantly outside equilibrium bonding regions and exhibit unphysical features. Moreover, they all incorrectly predict that two protons form a strongly-bound system, underscoring their failure to capture the basic Coulomb's law even when it is incorporated as an inductive bias. These shortcomings highlight the difficulty of achieving true physical generalization through scaling alone and suggest that current large-scale models risk acting solely as data-driven interpolators. Overall, the intersection of quantum chemistry and ML is in need of new strategies to make fast and accurate property predictions of novel molecules and materials.

## CODE AND DATA AVAILABILITY

Refer to this anonymized Zenodo repository for

- nuclear charges, nuclear coordinates, and atomization energies used as training data in our scaling experiments
- model checkpoints from our scaling experiments
- `Python` code to implement our scaling experiments
- `Python` code for utilizing the foundation machine-learned interatomic potentials

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
