# OpenReview forum: "Are neural scaling laws leading quantum chemistry astray?"
_ICLR.cc/2026/Conference — ICLR 2026 Conference Withdrawn Submission_

### Official Review · Reviewer_Zg9y · 2025-10-26

**Soundness:** 2
**Presentation:** 3
**Contribution:** 1
**Rating:** 2
**Confidence:** 5

**Summary:**

The authors show that machine-learned interatomic potentials (MLIPs) trained on molecular datasets (including large pre-trained models) fail to reproduce the $H_2$ dissociation curve.

**Strengths:**

The work is original in the sense that it investigates in depth a system that is usually not taken as a prototype out-of-distribution sample in other works (however, many other studies investigate the out-of-domain performance of MLIPs). The investigation is of good quality (although training models other than SchNet would have improved it) and the presentation is good.

**Weaknesses:**

As presented in its current form, this manuscript is of very little use to the ICLR community (or the computational chemistry community). The authors' expectations upon using MLIPs were probably too high and they point out a trivial out-of-distribution failure mode in the dissociation curve of $H_2$.

I am (briefly) going to highlight where I believe the manuscript went wrong:
- The authors probably expected the accuracy guarantees of neural scaling laws to apply to any type of data. They only apply *within the distribution of training samples*. No matter how much data we feed the NN, if the $H_2$ dissociation curve is not in it, we cannot expect improvement of its prediction. Many excellent pieces of work on the out-of-distribution behavior of MLIPs can be found throughout the literature.
- This is why much of the work in improving MLIPs has been focused on producing better datasets (i.e. better training distributions), which cover chemical compositions and/or configurations that might be of interest scientifically. Dataset design in this field must indeed improve and much work is devoted to it.
- Despite there being no accuracy guarantees on out-of-distribution predictions, uncertainty quantification can help flag pathological out-of-distribution cases for practitioners. Uncertainty quantification is starting to find its way into foundational MLIPs (e.g., MACE-MP, PET-MAD), and it partially addresses the problem the authors found.

I can see the authors turning the current manuscript into a thorough investigation of out-of-distribution behavior of foundational MLIPs for molecules, pointing out deficiencies of the OMol dataset and the lack of uncertainty quantification in the current largest pre-trained models (with experiments on smaller models providing it). For the time being, however, I do not see the manuscript as fit for publication in ICLR.

**Questions:**

At this stage, I have no questions for the authors.

---

### Official Review · Reviewer_dKu5 · 2025-10-28

**Soundness:** 3
**Presentation:** 3
**Contribution:** 2
**Rating:** 4
**Confidence:** 4

**Summary:**

This paper demonstrates that neural networks trained on quantum chemical data act as interpolators, failing to generalize to physical regimes (like bond dissociation) that are not well-represented in their training sets.

Through specially-designed experiments, the authors show:
* Models trained only on equilibrium-geometry data (GDB-9-G4(MP2)) fail to predict dissociation.
* Models trained on data that includes non-equilibrium structures (VQM24) perform qualitatively better.
* Large-scale foundation models (UMA, OMo125, etc.) also produce unphysical artifacts for simple diatomics and fail a basic Coulombic repulsion test for two bare protons, a physical regime they were not trained on.

The work concludes that scaling alone is insufficient for generalization, as models act as data-driven interpolators.

Overall, I find that this paper serves as an interesting read for the ML community as it provides some interesting experiment settings. However, I think the findings of this paper are a bit too intuitive and superficial, which is my major concern of this paper.

**Strengths:**

* The $H_2$ bond dissociation energy (BDE) curve test presented in the paper is interesting and can be leveraged for future model/dataset development.
* The paper is well written, using a language that is easy to follow and fun to read.

**Weaknesses:**

* The findings of this paper are a bit too intuitive. For neural networks, the lack of similar data almost certainly leads to prediction failures unless sufficient inductive biases are incorporated into the design -- in which case other data become "similar" to the data in question. Since the tested models do not have such inductive biases in them and seldom see data similar to those arise in the $H_2$ dissociation test, it is more than natural for them to fail.
* The findings of this paper are superficial. The paper finds the failures, but neither does it dig deeper into the phenomenon to find a reasonable cause, nor does it provide constructive suggestions on how to develop future models.
* The fact that the $H_2$ dissociation test is simple for quantum chemistry software does not imply it is simple for a deep learning model trained from unrelated data. This statement is also valid vice versa: deep learning models trained from small molecules can be easily applied to large molecules to obtain reasonable results, but it is not easy for quantum chemistry software to handle large molecules.

**Questions:**

* Could you provide a more detailed analysis on **why** the models fail on the tests?
* Could you provide some suggestions on **how** to improve future models on such tests? For example, is a small dataset enough for fixing the problem?

---

### Official Review · Reviewer_JF3e · 2025-10-29

**Soundness:** 3
**Presentation:** 3
**Contribution:** 2
**Rating:** 4
**Confidence:** 4

**Summary:**

Authors do the following: train SchNet models with different number of parameters on GDB-9-G4(MP2) and VQM24 datasets and then test how well such models reproduce the Hydrogen dissociation curve. Next, they take several foundational models (UMA-S, UMA-M, eSEN, AIMNet2, Orv v3) and test them on the dissociation of several diatomic molecules. Finally, they test foundational models on the system of two protons to see if the models “know” the Coulomb law. In general all the models struggle on most tasks.

**Strengths:**

Authors test several models on the dissociation of several diatomic molecule, which is a basic and important toy-problem for Force Fields. In general, the motivation and experiments are clear.

**Weaknesses:**

- The name of the paper seems to be misleading, authors test model size scaling laws for a very simple model (SchNet). Moreover, SchNet  is known to perform poorly on long-range and many-body interactions [1]. More complicated models must be tested.
- The implications of the results are not clear. How severe are the stated problems? Can we imply something for the more complex systems?
- All the models being GNNs were designed to work with relatively big systems, and small systems like $H_2$  do not give meaningful graph information. While it is still a drawback of the models, I am not convinced that it is as severe, as stated in the paper.
- Most importantly, the authors do not give practically any analysis of such behavior of the models and do not propose any solutions.

**Questions:**

- What is the motivation behind the choice of GDB-9-G4(MP2) and VQM24 datasets, while the first contains only relaxed conformations and the second relaxed and saddle points ? Why not choose a dataset like SPICE or nablaDFT with a good variability of system energies?
- The poor GNN performance on $H_2$ dissociation can be partially attributed to out-of-distribution test data as GDB-9-G4(MP2) and VQM24 datasets lack stretched or dissociating geometries. Can the introduction of more examples of stretched or breaking bonds to the training dataset improve the performance of GNN on dissociation of diatomic molecule? If so, what theory should be used for this data?
- As a major amount of foundational models were trained on DFT data and DFT itself does not reproduce the true dissociation curves, is it still a reasonable test?
- Do you think that switching to $\Delta$ ML may fix some of the problems?
- While it is not necessary, figure 1 lacks the curves for G4(MP2) level of theory.

[1] Esders, M., Schnake, T., Lederer, J., Kabylda, A., Montavon, G., Tkatchenko, A., & Müller, K. R. (2025). Analyzing atomic interactions in molecules as learned by neural networks. Journal of Chemical Theory and Computation, 21(2), 714-729.

---

### Official Review · Reviewer_xaPM · 2025-10-31

**Soundness:** 2
**Presentation:** 2
**Contribution:** 3
**Rating:** 4
**Confidence:** 4

**Summary:**

This paper investigates the limits of machine-learning interatomic potentials (MLIPs) trained on large datasets by evaluating their ability to predict bond dissociation curves for ($H_2$) and other simple diatomic molecules. The authors show that simply scaling model size and dataset size does not resolve errors in this fundamental extrapolation task, and that state-of-the-art foundation models trained on diverse chemical datasets still exhibit unphysical behavior in the dissociation curves. In addition, they find that Coulomb repulsion between bare protons is not modeled correctly. Based on these observations, the authors argue that scaling MLIPs alone is insufficient to learn fundamental chemistry.

Note: An LLM (ChatGPT 5) was used in this review to expand preliminary notes into the full text. It did not contribute suggestions or questions.

**Strengths:**

- The paper highlights a severe and important failure mode for MLIPs, demonstrating that even large models trained on massive datasets can fail on extremely basic physical scenarios.
- It raises a timely and meaningful question about "blind" scaling of model size and data, encouraging consideration of when scaling does and does not lead to better generalization.
- The study shows that including explicit long-range corrections does not guarantee correct Coulombic behavior for simple proton–proton systems, which is notable and worth deeper examination.
- The paper serves as a compact and well-written introduction to quantum chemistry for ML practitioners.

**Weaknesses:**

- The overall framing of the work appears to me to be overstated. The experiments show that scaling a specific architecture does not resolve certain failures and that foundation models exhibit some errors, but they do not demonstrate that scaling as a paradigm is misguided. The argument seems to assume a particular interpretation of the scaling hypothesis -- that models will learn quantum chemistry principles purely from scale -- which should be explicitly acknowledged. My understanding of the prevailing view in the field is that scaling improves the chance that physically reasonable (or practically relevant) structures lie near the training distribution, not that scaling alone "teaches" fundamental physics.
- The practical impact of the reported shortcomings is unclear. As the authors note, standard KS-DFT also does not correctly model the ($H_2$) dissociation limit, yet DFT is widely used in production. The paper should explain why MLIPs should be held to a stricter standard, or argue concretely why this matters in practice.
- Similarly, the relevance of the bare-proton example is not clearly justified. It is a pathological case that appears very far from the types of molecular configurations encountered in practice.
- The scaling experiment relies on SchNet, which is no longer state of the art. In addition, the models are trained only on energies, which is also not typical for MLIPs. While this may not change the qualitative conclusion, the choice weakens the strength and modern relevance of the results.
- The paper mixes results and training data from multiple levels of electronic-structure theory, making it challenging to understand what accuracy and behavior one should expect from different models relative to their training sources.

**Questions:**

- Please expand the discussion around "right for the right reasons". If MLIPs truly emulated first-principles quantum chemistry, they would incur similar computational cost. Instead, models necessarily use learned shortcuts. Why should they be expected to generalize correctly far outside their training data, especially if they do not include explicit physical priors such as a repulsive term? The common foundation-model intuition is that interesting, practically relevant regions of chemical space form a much smaller manifold than all of chemical space, and the goal is to approximate behavior well there -- not necessarily on every possible configuration.
- Please discuss the practical relevance of the findings. What applications would be directly impacted by these failures? Are there cases where incorrect dissociation curves could meaningfully mislead molecular-design workflows or observables in molecular dynamics?
- I encourage the authors to repeat the scaling experiment using a more modern architecture and training protocol that more closely reflects current state of the art, including force training.
- A table summarizing each model, its training dataset, level of theory, and expected fidelity for dissociation-curve prediction would significantly improve clarity for readers.
- Most modern MLIPs apply a finite interaction cutoff. Was this considered or controlled for in the experiments? If so, please clarify; if not, this could be an important factor affecting long-range behavior. Note that there has been much recent work on including long-range effects in MLIPs, which could be considered here (Frank et al., arXiv:2412.08541; Rumiantsev et al., arXiv:2507.19382; Caruso et al., arXiv:2502.13797; Zhadanov et al., ICML 2025 https://openreview.net/forum?id=MrphqqwnKv; Cheng, DOI:10.1038/s41524-025-01577-7, Unke et al., DOI:10.1038/s41467-021-27504-0, etc.)
- The plots are difficult to parse. Please consider improved figure readability and possibly zoomed-in subplots to highlight key behaviors around equilibrium and long-range limits.
- It would be valuable to speculate on why the observed failure modes arise. Are they likely due to missing physical priors, insufficient off-equilibrium training data, architectural limitations, or something else?
- Please comment on Chiang et al. (arXiv:2509.20630, https://huggingface.co/spaces/atomind/mlip-arena). Their supplementary Figure S1 also contains diatomic dissociation curves, and some models appear to behave reasonably there while others do not. How do those results relate to your findings?

---

### Note · Authors · 2025-12-05

I have read and agree with the venue's withdrawal policy on behalf of myself and my co-authors.